## Research Article

**Key words:**
Metadynamics; multi-target pain modulator; pain treatment

**Author for correspondence:**
*William A. Goddard III,
E-mail: wag@caltech.edu

**CAMBRIDGE UNIVERSITY PRESS**

# The G protein-first activation mechanism of opioid receptors by Gi protein and agonists

Amirhossein Mafi [ID], Soo-Kyung Kim [ID] and William A. Goddard III* [ID]

Materials and Process Simulation Center (139-74), California Institute of Technology, Pasadena, CA 91125, USA

## Abstract

We report the G protein-first mechanism for activation of G protein-coupled receptors (GPCR) for the three closest subtypes of the opioid receptors (OR), μOR, κOR and δOR. We find that they couple to the inactive Gi protein-bound guanosine diphosphate (GDP) *prior* to agonist binding. The inactive Gi protein forms anchors to the intracellular loops of the *inactive* apo-μOR, apo-κOR and apo-δOR, inducing opening of the cytoplasmic region to form a pre-activated state that holds Gi protein in place until agonist binds. Then, agonist binds to μOR, κOR and δOR already complexed with Gi protein, to trigger the Gαi to open up the tightly coupled GDP binding site, making GDP accessible for GTP exchange, an essential step for Gi signalling. We show that the agonist alone *cannot* open the intracellular region of μOR and κOR, requiring Gi protein to open the cytoplasmic region by itself. We consider that this G protein-first mechanism may apply to activation of other Class A GPCRs. However, for δOR, agonist binding can open up the intracellular region to encourage Gi protein recruitment. Thus, activation of Gi protein mediated by δOR favourably may proceed with either ligand-first or G protein-first activation mechanisms.

## Introduction

Chronic pain treatment is a major clinical challenge because most opioid analgesics such as morphine are associated with the side effects that hinder their application. Thus, current medications do not provide sufficient pain relief. As a result, there is a great need to develop new pain therapeutics that attenuate the pain signals without the side effects. The primary target of morphine and other clinical opioid analgesics is the μ-opioid receptor, μOR (Pasternak and Pan, 2013), a G protein-coupled receptor (GPCR) that stimulates analgesic activity through signalling via the adenylyl cyclase-inhibitory family of G protein, Gi/o (Al-Hasani and Bruchas, 2011). Concomitantly, the opioid analgesics can also act on κ-opioid receptor (κOR) and δ-opioid receptor (δOR), which altogether constitute three closest subtypes of opioid receptors that share 70% identity in their transmembrane (TM) domains (Waldhoer *et al.,* 2004). The negative side effects associated with prescription opioids stem from the activation of μOR (Matthes *et al.,* 1996; Zadina *et al.,* 1997; Liu *et al.,* 2011) and δOR (Clapp *et al.,* 1998; Jutkiewicz *et al.,* 2006), whereas therapeutics activating κOR confer analgesia in both human and animals (Schmauss and Yaksh, 1984; Nakazawa *et al.,* 1985; Pande *et al.,* 1996) with fewer side effects (Pan, 1998; Bruchas and Roth, 2016). Therefore, a multi-target pain modulator, that agonises κOR while simultaneously antagonising μOR and δOR, offers a promising approach to dramatically reduce neuropathic pain as well as avoiding the common side effects. To develop new analgesics with high efficacy but reduced side effects, it is critical to understand the activation mechanism underlying the choreography among μOR/κOR/δOR, Gi protein and agonists.

Generally, it is assumed that binding of agonists to inactive GPCRs shifts the equilibrium towards an activated conformation of the receptors (Clark, 1926; Karlin, 1967). The activation of GPCRs is associated with a large opening between the cytoplasmic ends of TM6 and TM3, dramatically expanding the spacing in the intracellular region of the GPCR (Hilger *et al.,* 2020). This expansion facilitates recruiting and activating G protein-bound guanosine diphosphate (GDP), which is later exchanged with guanosine triphosphate (GTP), to mediate rapid signalling (dissociation of Gα and Gβγ subunits into free active subunits) (Gilman, 1987; Bourne, 1997; Cabrera-Vera *et al.,* 2003). In the ligand-first mechanism of activation, it is assumed that recruitment of the G protein depends greatly on random collisions between the activated receptor-bound agonist and the G protein, which is controlled by diffusion of the G protein (Orly and Schramm, 1976; Tolkovsky and Levitzki, 1978). Therefore, activation of the GPCR first by a ligand and then coupling to the G protein are critical steps towards the signalling. Typically, a strong coupling between the cytoplasmic end of TM3 and TM6 stabilises the inactive state of Class A GPCRs (Sheikh *et al.,* 1996; Ballesteros *et al.,* 2001), which inhibits TM6 outward movements. Thus, disruption and breaking of this coupling is a critical event in the activation of GPCRs (Ballesteros *et al.,* 2001; Yao *et al.,* 2006; Kobilka, 2007). In the process of activation, the G protein undergoes a significant separation of the α-helical (AH) domain of the Gα subunit from

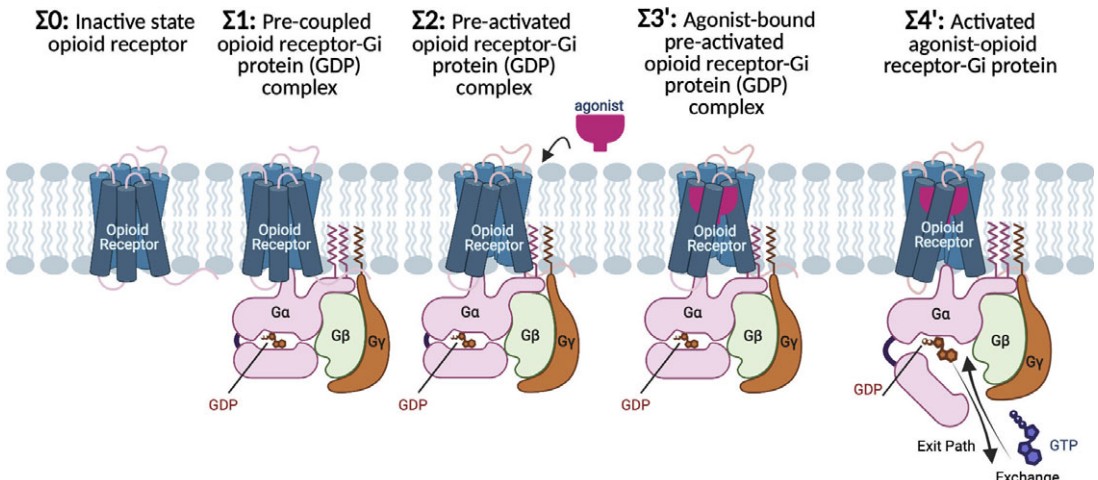

**Fig. 1.** G protein-first mechanism of activation for opioid receptors and their cognate Gi protein. Σ0: In the absence of ligand and Gi protein, the opioid receptors adopt the inactive conformation, featuring a tight hydrogen bond between the cytosolic ends of TM3 and TM6 that keeps the cytoplasmic region tightly closed. Σ1: Before agonist binding, the inactive Gi protein tightly bound to GDP couples to inactive opioid receptor, to form a pre-coupled opioid receptor-Gi (GDP) complex. Σ2: Interactions between inactive opioid receptor and inactive Gi (GDP) leads to breaking the TM3-TM6 hydrogen bond and opening the cytoplasmic region of the receptors to accommodate the Gi protein. As a result, the pre-activated state (Σ2) emerges, which remains at this resting state until an agonist binds the receptor. Σ3′: agonist bound to the pre-activated state induces the Gi (GDP) to be activated. Activation of the Gi protein is associated with a remarkable opening in the cleft between AH and Ras-like domains of Gα, providing an exit path for GDP release or exchange with a GTP. Σ4′: Upon GDP release of exchange, the agonist-opioid receptor-Gi protein evolves to its fully active state.

the RAS-like domain, which opens the nucleotide binding pocket to facilitate the exchange of GDP for a GTP nucleotide (Sprang, 1997; Oldham and Hamm, 2008).

There remains considerable uncertainty about the mechanism by which agonists induce GPCRs to activate their cognate G proteins. Some GPCRs (Dror *et al.,* 2011; Rasmussen *et al.,* 2011; Nygaard *et al.,* 2013; Manglik *et al.,* 2015; Kato *et al.,* 2019), particularly μOR (Sounier *et al.,* 2015), feature a weak allosteric coupling between the ligand-binding pocket and the G protein coupling interface such that the agonist alone cannot stabilise the expanded cytoplasmic region of the GPCR in the active state conformation. This is in stark contrast to the assumption that ligand binding shifts the equilibrium towards the active conformation of receptors. Moreover, several GPCRs, particularly μOR, exhibit constitutive activity in the absence of ligand (Liu *et al.,* 2001; Okude *et al.,* 2015; Sena *et al.,* 2017), suggesting that activation of G protein by GPCRs need not always depend on the presence of an agonist. Envisioned random collisions between between G protein and receptors in the ligand-first mechanism of activation are comparatively slow given that cells constitute various receptors, G protein subunits and other downstream effectors such as arrestin, all of which may compete with the G protein to couple to the receptor. Hence, the ligand-first mechanism of activation cannot adequately describe how G proteins are rapidly activated (Gilman, 1987; Bourne, 1997; Cabrera-Vera *et al.,* 2003) by activated receptors.

These inconsistencies invoked an opposite hypothesis in which G proteins prior to ligand binding can directly interact with GPCRs to make a pre-coupled complex (Nobles *et al.,* 2005; Galés *et al.,* 2006; Ayoub *et al.,* 2007; Qin *et al.,* 2011; Kilander *et al.,* 2014; Andressen *et al.,* 2018). Interestingly, it was shown that the pre-coupled complex between the inactive G protein and inactive GPCR eventually leads to rapid G protein activation after the agonist binds to the receptor-G protein complex (Qin *et al.,* 2011). Although the emergence of a pre-coupled G protein-GPCR complex has been observed previously (Nobles *et al.,* 2005; Galés *et al.,* 2006; Ayoub *et al.,* 2007; Qin *et al.,* 2011; Kilander *et al.,* 2014; Andressen *et al.,* 2018), the detailed molecular mechanism

by which both GPCR and G protein are activated through the G protein-first mechanism of activation remains not understood.

In this paper, we investigate the G protein-first paradigm (Fig. 1) using long-scale (~21 μs total) molecular dynamics (MD) simulations to follow the sequence of structural and energetic steps involved in activation of both the opioid receptors and the Gi protein. The activation process goes through several metastable states in which the GPCR structure undergoes various structural changes that define the important events during activation. Some of these metastable states may be separated by high energy barriers that may take microseconds or longer. Thus, we used meta-molecular dynamics (metaMD) simulations (Barducci *et al.,* 2008), in which relevant collective variables describing the slow degrees of freedom are biased to encourage the system to explore large regions of conformational phase space in much reduced time. We followed two important slow degrees of freedoms associated with activation of opioid receptors and Gi protein.

(i)     Opening the strong coupling between TM3 and TM6 in the inactive opioid receptors (μOR/κOR/δOR). The disruption and breaking of this coupling are critical events in activation of Class A GPCRs (Ballesteros *et al.,* 2001; Yao *et al.,* 2006; Kobilka, 2007).

(ii)    Opening the tight Gαi subunit coupled to GDP to an open form that enables the signalling arising from GDP-GTP exchange (Sprang, 1997; Oldham and Hamm, 2008).

We report here the discovery that prior to binding of an agonist to the opioid receptors (μOR/κOR/δOR), the cognate Gi protein forms salt bridge anchors to all three intracellular loops (ICL) of the inactive opioid receptor, aligning the Gα5 helix to extend partially into the receptor to form a pre-activated complex. For the inactive conformation of opioid receptors, the conserved $R^{3.50}$ (part of DRY motif in Class A GPCRs) establishes a polar interaction with the conserved $T^{6.34}$ that locks the intracellular region closed. In the pre-activated (μOR/κOR/δOR)-Gi protein complex, we find that the terminal carboxylate of the Gα5 helix forms a salt bridge with $R^{3.50}$,

weakening the coupling between TM3 and TM6, which initiates expansion of the cytoplasmic GPCR region to accommodate the Gα5 helix. This pre-activated state is stable until agonist binds to this pre-activated (μOR/κOR/δOR)-Gi protein complex to induce the Gαi subunit to undergo a dramatic opening at the GDP binding site by ~16.0 to 24.0 Å. This exposes the GDP to water, making it susceptible to nucleotide exchange with GTP. Thus, binding of agonist converts the pre-activated opioid receptor-Gi protein complex to the fully activated complex. This discovery provides a new target for the design of improved selective multi-target pain modulators.

## Results

### Activated state of opioid receptors–agonist-Gi complex

The transducing signalling for GPCRs requires communications from the ligand-binding site in the extracellular portion to the intracellular domain of the receptor where the cognate G protein is recruited. During the activation process, the receptor conformation evolves from an inactive state (denoted as Σ0) to a fully activated state (denoted as Σ4′). To obtain the structure for the human opioid receptors bound with the full Gi protein and agonists (Σ4′), we started with the 3.5 Å resolution Cryo-electron microscopy (Cryo-EM) structure (Koehl *et al.,* 2018) of mouse μOR bound to DAMGO and the nucleotide-free Gi protein. Unfortunately, the Cryo-EM μOR-Gi protein structure *did neither resolve* the whole AH domain of Gαi subunit (missing residues 56–181 and 234–240) nor did it resolve the full side chains for five residues important for μOR-Gi protein coupling (including E28, E308 and E318 in the Gαi subunit, D312 in the Gβ subunit and K100$^{ICL1}$ in the μOR).

Therefore, we built in the missing 133 residues of the AH domain from the active state complex of the human rhodopsin and Gi protein (PDB ID: 6CMO) and modelled in the five missing side chains. Subsequently, we immersed the resulting construct in the lipid bilayer, water and ions and carried out an aggregate of ~450 ns MD simulations with restraints on the Cryo-EM backbone atoms to ensure that the shape of proteins not be disturbed as the missing added segments are relaxed.

Strikingly, we find that Gi protein couples to μOR by forming strong salt bridge anchors to each of three ICLs (Fig. 2a–h).

- Our optimised complex indicates that the Gβ subunit binds directly to ICL1 by forming a strong $(-1.7 \pm 0.3$ kcal mol$^{-1})$ salt bridge: D312$^{Gβ}$-K98$^{ICL1}$ (Fig. 2b).
- The Gαi subunit interacts with the ICL2 and the cytosolic end of TM4 by making two pairs of salt bridges: R32$^{GαiN-β1\ loop}$-D177$^{ICL2}$ and E28$^{GαiN}$-R182$^{4.40}$ (Fig. 2c). The superscripts are Ballesteros–Weinstein numbering for GPCRs (Pándy-Szekeres *et al.,* 2017). To assess the strength of the salt bridge between R32$^{GαN-β1\ loop}$-D177$^{ICL2}$, we carried out a ~1.6 μs metaMD simulation (Fig. 2f) to find that forming this salt bridge substantially decreases the energy by ~3 kcal mol$^{-1}$.
- Similarly, the Ras-like domain of Gαi couples to the ICL3 and the cytoplasmic end of TM6 by making two pairs of salt bridges: E318$^{Gαi-α4-β6\ loop}$-R263$^{ICL3}$ and E318$^{Gαi-α4-β6\ loop}$-K271$^{6.26}$ (Fig. 2d). Our free energy calculations reveal high affinity between these pairs of salt bridges and $-2.1$ with $-1.4$ kcal mol$^{-1}$ (Fig. 2g,h).

Interestingly, none of these ionic anchors were identified in the Cryo-EM structure (Koehl *et al.,* 2018) − because E28, E308 and E318 in the Gαi subunit and D312 in Gβ subunit were not fully resolved.

Our MD simulations indicate that the activated Gαi-α5 helix engages in extensive polar interactions with μOR (Supplementary Fig. S2). Overall, we located 18 polar interactions of which only 5 were reported in the Cryo-EM structure. The other 13 polar interactions emerge readily while the backbone of the Cryo-EM construct remains fixed. Forming these salt bridges and hydrogen bonds leads to a final structure with root mean square deviation (RMSD) = 1.3 Å, well within the experimental resolution. Fig. 2i compares the optimised and Cryo-EM complexes. Interestingly, the calculated density map from MD has a better correlation, ~0.9, with the protein coordinates resolved by Cryo-EM, suggesting that our optimised structure can be considered as an experimental structure enhanced to achieve the atomic resolution of the full Gi protein-μOR DAMGO complex. We used the μOR-Gi complex as a template for applying various computational methods to obtain the fully active structure of the other opioid receptors bound to agonist and Gi protein.

We used the active conformation of mouse μOR (Huang *et al.,* 2015) as a template for GEnSeMBLE (Bray *et al.,* 2014) complete sampling predictions to obtain the 3D structure of human μOR. The Cryo-EM structure contained the DAMGO agonist peptide, but we used morphine, a clinical agonist. Thus, we employed the DarwinDock (Griffith, 2017) complete sampling method to predict the binding site of morphine to the human μOR. The resulting human μOR-morphine complex was superimposed onto the optimised mouse μOR-Gi complex to obtain the fully active state construct. Then, we equilibrated the resulting construct by performing a ~1 μs MD simulation (Fig. 3a), leading to the Σ4′ fully activated structure. We find that the Gi protein interfaces the human μOR by forming salt bridge anchors to ICL1, ICL2 and the cytoplasmic end of TM6.

Our analysis shows that the Gβ subunit makes a direct and stable ionic contact from D312$^{Gβ}$ to K100$^{ICL1}$ (Fig. 3b,h). Interestingly, the Cryo-EM structures of the activated glucagon-like peptide-1 receptor complexed with Gs protein find that the same D312 makes a salt bridge with H171 in the ICL1 of GLP1 (Zhang *et al.,* 2017; Liang *et al.,* 2018). In addition, the Cryo-EM structure of the adenosine A$_{2A}$ receptor bound to a mini Gs protein (García-Nafría *et al.,* 2018) also finds that the Gβ subunit makes polar contacts to ICL1, showing the significant role of Gβ in modulating G protein coupling.

We find that the Gαi subunit also makes a charge–charge contact with ICL2: R32$^{GαiN-β1\ loop}$ to D179$^{ICL2}$ (Fig. 3c,g). This anchor coordinates R181$^{ICL2}$ to involve a network of polar interactions (Fig. 3d), playing a crucial role in stabilising the complex. In fact, the R181C mutation inhibits transduction signalling *in vitro* (Ravindranathan *et al.,* 2009) causing patients to become insensitive to morphine (Skorpen *et al.,* 2016). The third set of ionic anchor emerges between E318$^{Gαi-α4-β6\ loop}$ and K273$^{6.26}$ that tightly couples the Ras-like domain and the Gαi-α5 helix to the cytoplasmic region of the μOR, stabilising the active position of Gαi-α5 helix (Fig. 3e,i).

To eliminate the possibility that our discovery of ionic anchors might have resulted from our choice of force fields, Amber 14 (Dickson *et al.,* 2014), we performed two independent 1 μs of MD simulations (Supplementary Fig. S3) using the Charmm36m (Huang *et al.,* 2017) and OPLS (Robertson *et al.,* 2015) force fields. We find that the optimised complex obtained from all three of these well-validated force fields features ionic anchors between the Gi protein and the intracellular region of the μOR, confirming that the

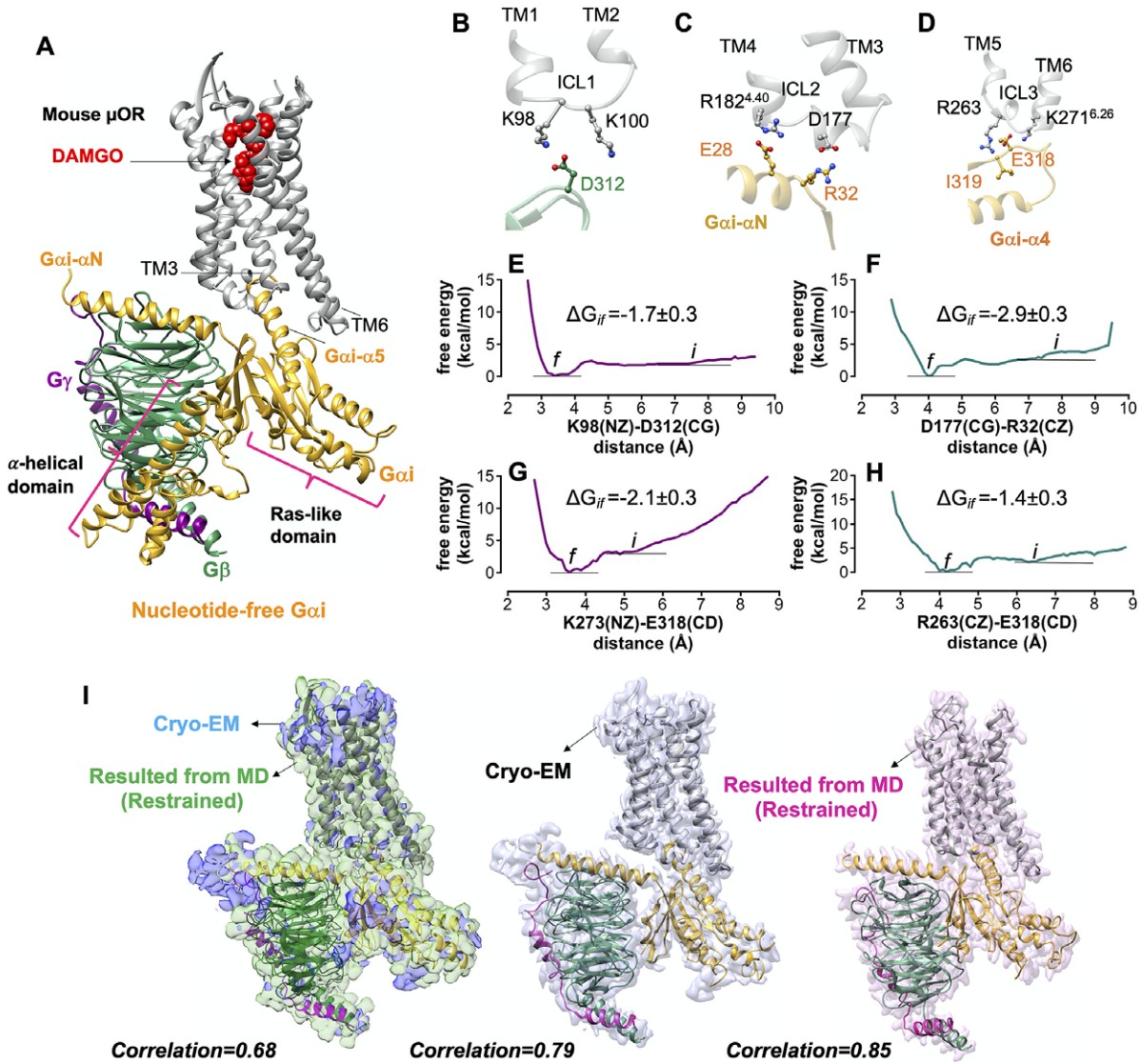

**Fig. 2.** Gi protein binds the mouse μOR by forming ionic anchors to each of three ICLs. (*A*) The energy minimised mouse μORGi complex derived from ~450 ns MD and ~1.8 μs metaMD simulations. After ~450 ns MD simulation the RMSD = 1.3 Å from the original Cryo-EM structure. (*B*) The salt bridge anchor from the Gβ subunit to the ICL1. (*C*) Salt bridge anchors from the Gαi subunit to ICL2 and cytoplasmic end of TM4. (*D*) Salt bridge and hydrogen bond anchors from the Gαi subunit to the ICL3 and to the cytoplasmic end of TM6. Binding free energy between ionic anchors from metaMD. (*E*) K98(NZ)-D312(CG) salt bridge coupling Gβ to ICL1, (*F*) D177(CG)-R32(CZ) salt bridge coupling Gα to ICL2, (*G*) K273(NZ)-E318(CD) and (*H*) R263(CZ)-E318(CD) salt bridges coupling Gα to ICL3 and TM6.) Comparison of the optimised complexes from MD simulations with the Cryo-EM structure. Alignment of the density-map obtained by MD simulation with the backbone atoms restrained to the Cryo-EM density map (left). Here, the explicit structure is the snapshot at ~450 ns of MD simulation. (Middle): Alignment of the Cryo-EM structure (PDB ID: 6ddf) to the Cryo-EM density map. (Right): Alignment of the Cryo-EM structure (PDB ID: 6ddf) to the density map obtained from MD simulation with the backbone atoms restrained. Interestingly, our optimised density map has a better correlation with the Cryo-EM structure (PDB ID: 6ddf). Thus, our refined mouse structure can be considered as an experimental structure enhanced to achieve the atomic resolution of the full Gi-μOR-agonist complex. The weighted averages and the standard deviations were calculated for the converged period between the initial configuration before metaMD '*i*' and the final conformation '*f*' after metaMD calculations (Supplementary Fig. S1).

emergence of salt bridge anchors between μOR–Gi protein is not dependent to the choice of force field.

To determine whether the ionic anchors coupling the Gi protein to μOR, are restricted to μOR, we predicted the fully active state of κOR-MP1104-Gi (Mafi *et al.*, 2020) and δOR-DPI-287-Gi complexes. To predict these complexes, we followed our recent procedure (Grisshammer, 2020; Mafi *et al.*, 2020) in which we removed the mimetic G protein nanobody from the active conformation of κOR (PDB ID: 6B73) (Che *et al.*, 2018) and δOR (PDB ID: 6PT2) (Claff *et al.*, 2019), and replaced it with our optimised Gi protein bound to the mouse μOR. Subsequently, we relaxed the resulting constructs by performing MD simulations to obtain the optimised

active state complexes (full details provided in the Supplementary Information). Our analysis shows that Gi protein makes similar salt bridge anchors to ICL1, ICL2 and the cytosolic end of TM6 in the fully activated complex, Σ4′ (Fig. 4) for both κOR and δOR. This shows that the emergence of these ionic anchors is a common feature of opioid receptor subtypes. This finding suggests that salt bridge anchors between Gi protein and opioid receptors play essential roles for activation and consequently G protein signalling. Indeed, we find that these three anchors serve as a tripod orienting and positioning the Gi protein so that its Gαi-α5 helix is lined up for insertion into the μOR to establish the extensive interactions that stabilise the active state complex.

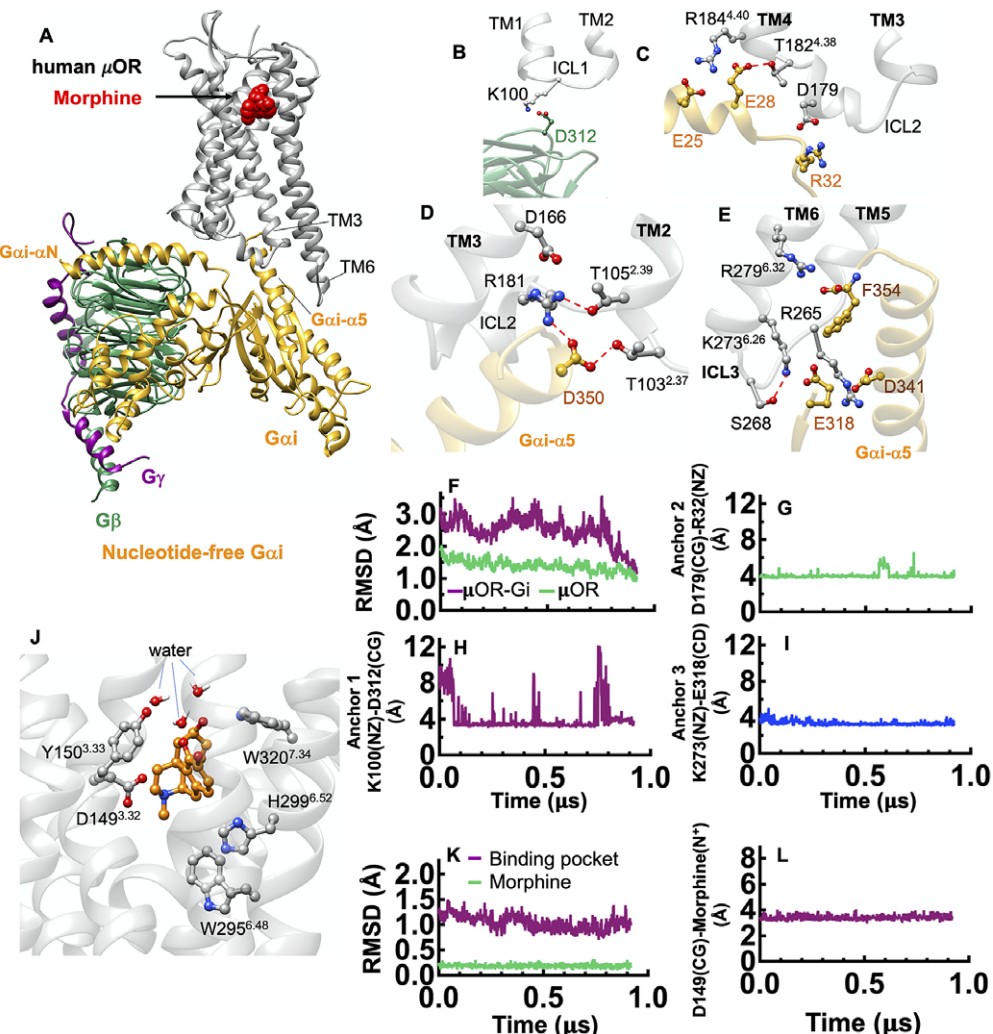

**Fig. 3.** Gi protein binds the human μOR by forming ionic anchors to ICL1, ICL2 and the cytoplasmic end of TM6. (*A*) Structure of the human μOR–Gi protein complex derived from a ~950 ns MD simulation using Amber14. (*B*) The ionic anchor from the Gβ subunit to the ICL1. (*C*) Salt bridge anchors from the Gαi subunit to ICL2 and to the cytoplasmic end of TM4. (*D*) The network of polar interactions between ICL2 and the Gαi-α5 helix and (*E*) ionic anchors from the Gαi subunit to the ICL3 and the cytosolic end of TM6. (*F*) RMSD variation of the complex with time. Here, the RMSD calculated for the backbone atoms of the whole structures over the simulation and compared to the final snapshot. (*G-I*) Variation of the salt bridge anchors between Gi protein-μOR with time. The dotted red lines indicate hydrogen bonding. (*J*) Human μOR binding pocket after ~950 ns of MD simulation. The salt bridge between D149(CG) and morphine (the protonated N atom), locks morphine in the orthosteric binding pocket. (*K*) RMSD variation for the binding pocket and morphine with time. (*L*) The key salt bridge interaction between D149$^{3.32}$ and the morphine protonated N atom (the protonated N atom) that holds morphine in tight contact with the human μOR.

## *Mechanism of G protein activation prior to agonist binding*

Prior to the ligand binding, we hypothesise that Gi protein has sufficient time to couple to the opioid receptors to form a pre-coupled state. To examine whether Gi protein can spontaneously couple to each of opioid receptors, we followed our proposed mechanism of G protein activation:

1. The apo-opioid receptor initially exhibits a tight cytoplasmic region due to a polar interaction between R$^{3.50}$ (part of DRY motif) and the conserved T$^{6.34}$ of the opioid receptors. This coupling constitutes the slowest degree of freedom for the activation of opioid receptors. The disruption and breaking of this coupling are believed to be critical events in activating GPCRs (Kobilka, 2007).

2. Prior to agonist binding, we find that the Gi protein interfaces with the apo-opioid receptors by making salt bridge anchors to the three ICLs, thereby aligning the Gα-α5 helix such that

its terminal carboxylate (F354) is able to form a salt bridge with R$^{3.50}$.

3. Formation of salt bridge: F354-R$^{3.50}$ breaks the coupling between TM3-TM6 [R$^{3.50}$-T$^{6.34}$], which consequently opens up the cytoplasmic region of the opioid receptor to facilitate insertion of the Gα-α5 helix partly into the receptor core, forming the pre-activated state (Σ2) between Gi protein and apo-opioid receptors. Next, an agonist binds to the pre-acti-vated complex (forming Σ3′) that subsequently cause the Gα to open up the Ras-like and AH domains binding to the GDP protein, leading to the fully active state (Σ4′) with GDP bound only to the Ras-like domain.

We formed a model (the full details provided in the Supplementary Information) of the pre-coupled complex (denoted as Σ1) between inactive human μOR (denoted as Σ0) and the tight Gi protein bound to GDP (Fig. 5a). The inactive human μOR initially features

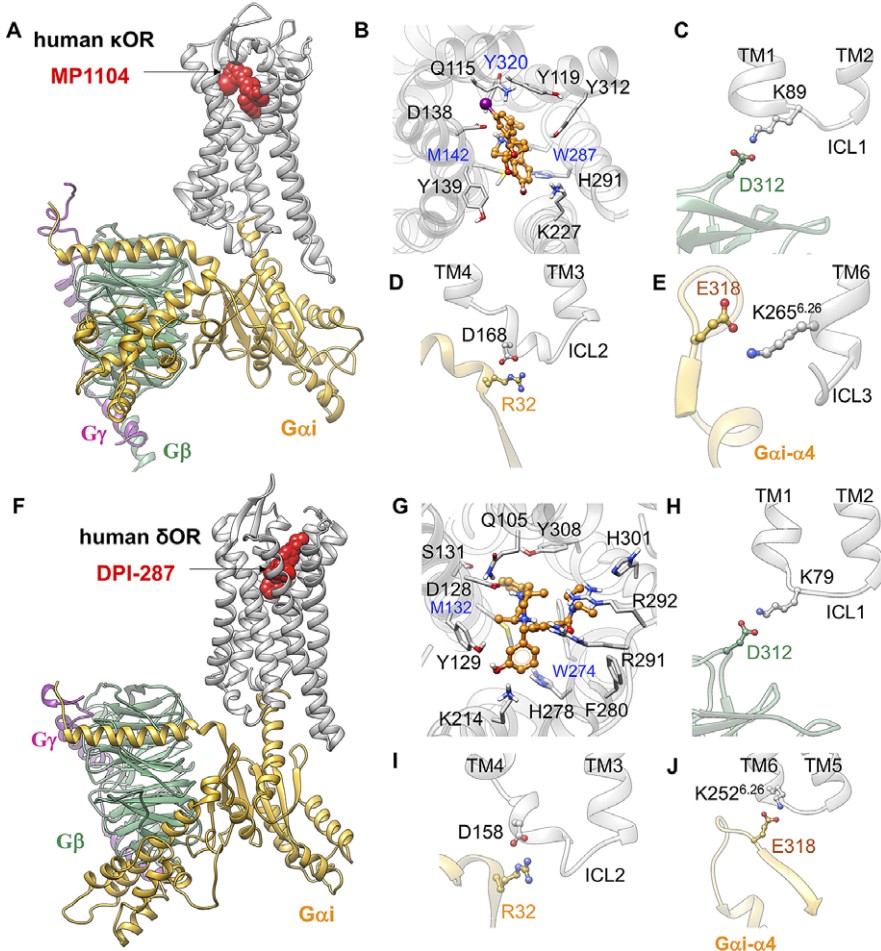

**Fig. 4.** Gi protein binds the human κOR and δOR by forming ionic anchors to ICL1, ICL2 and the cytoplasmic end of TM6. (*A*) Structure of the human κOR–Gi protein-MP1104 obtained from MD simulation. (*B*) MP1104 binding pocket, where the salt bridge between D138³·³² and MP1104 (the protonated N atom) holds MP1104 in tight contact with the human κOR. (*C*) The ionic anchor from the Gβ subunit to the ICL1. (*D*) The salt bridge anchor from the Gαi subunit to ICL2. (*E*) The ionic anchor from the Gαi subunit to the cytosolic end of TM6. Here, (*A-E*) adapted from figs 3 and 4 of Mafi *et al.* (2020). (*F*) Structure of the human δOR–Gi protein-DPI-287 obtained from ~300 ns MD simulation using Charmm36m force field. (*G*) DPI-287 binding pocket, where the salt bridge between D128³·³² and DPI-287 (the protonated N atom) locks DPI-287 in the orthosteric binding pocket of the human δOR. (*H*) The ionic anchor from the Gβ subunit to the ICL1. (*I*) The salt bridge anchor from the Gαi subunit to ICL2. (*J*) The ionic anchor from the Gαi subunit to the cytosolic end of TM6.

a strong polar interaction between R167³·⁵⁰ and T281⁶·³⁴. In the Σ1 state, Gi protein was placed in close enough proximity of the inactive μOR that it could form ionic anchors to ICL1, ICL2, ICL3 and the cytosolic end of TM6 (Fig. 5*a*). The inactive Gαi subunit is bound tightly to the GDP (Lambright *et al.,* 1996) coupling the helical and Ras-like domains. The starting orientation and position of the Gαi-α5 C-terminal helix is well beneath the intracellular region of inactive μOR, to avoid steric clashes between Gi protein and inactive receptor. Subsequently, we allowed the pre-coupled complex to find the optimum position and orientation of Gαi-α by performing a ~1 μs metaMD simulation. Our free energy calculations (Fig. 5*b,c*) reveal that the terminal carboxylate of Gαi subunit, F354, moves ~6 Å to make a salt bridge with R167³·⁵⁰ (~−2 kcal mol⁻¹). In fact, the salt bridge: F354-R167³·⁵⁰ weakens the intrinsic polar interaction between R167³·⁵⁰ and T281⁶·³⁴, which opens ultimately to ~6 Å. Upon breaking this hydrogen bond, T281⁶·³⁴ rotates towards TM5, facilitating the penetration of the Gαi-α5 helix into the receptor core (Fig. 5*c,d*). Our free energy calculations indicate that opening this TM3-TM6 coupling prior to agonist binding is spontaneous, substantially decreasing the energy by ~−2.4 kcal mol⁻¹ (Fig. 5*c*). We denote this as the pre-activated state (Σ2 state). The association of μOR with its cognate Gi protein

prior to the agonist binding is consistent with the constitutive activity that μOR exhibits in its apo form (Liu *et al.,* 2001; Okude *et al.,* 2015; Sena *et al.,* 2017).

There remains a possibility that the rigid-body orientation of Gi protein could be very different in the pre-coupled state from that in the fully active complex. To eliminate the possibility that the specific rigid-body orientation used in the pre-coupled state (Σ1) is solely responsible for opening the TM3-TM6 coupling, we carried out an independent ~1 μs metaMD free energy calculation in which we included only the Gαi-α5 peptide (the last 21 residues: ³³⁴F-F³⁵⁴) placed in close proximity to the inactive μOR (Fig. 5*e–i*). The increased degrees of freedom for the Gαi-α5 peptide enabled us to explore various positions and orientations (Supplementary Fig. S12), which would emerge from various orientations of whole Gi protein in complex with the μOR. Our analysis shows that prior to ligand binding, a charge–charge contact from the terminal carboxylate, F354, to R167³·⁵⁰ (Fig. 5*h*) contributes to opening the strong coupling between R167³·⁵⁰-T281⁶·³⁴. After breaking the TM3-TM6 coupling, the Gαi-α5 peptide penetrates deep into the core of μOR to establish a hydrogen bond with N276⁶·²⁹. These calculations confirm that the formation of the pre-activated state between Gi protein and μOR is not an artefact resulting from a

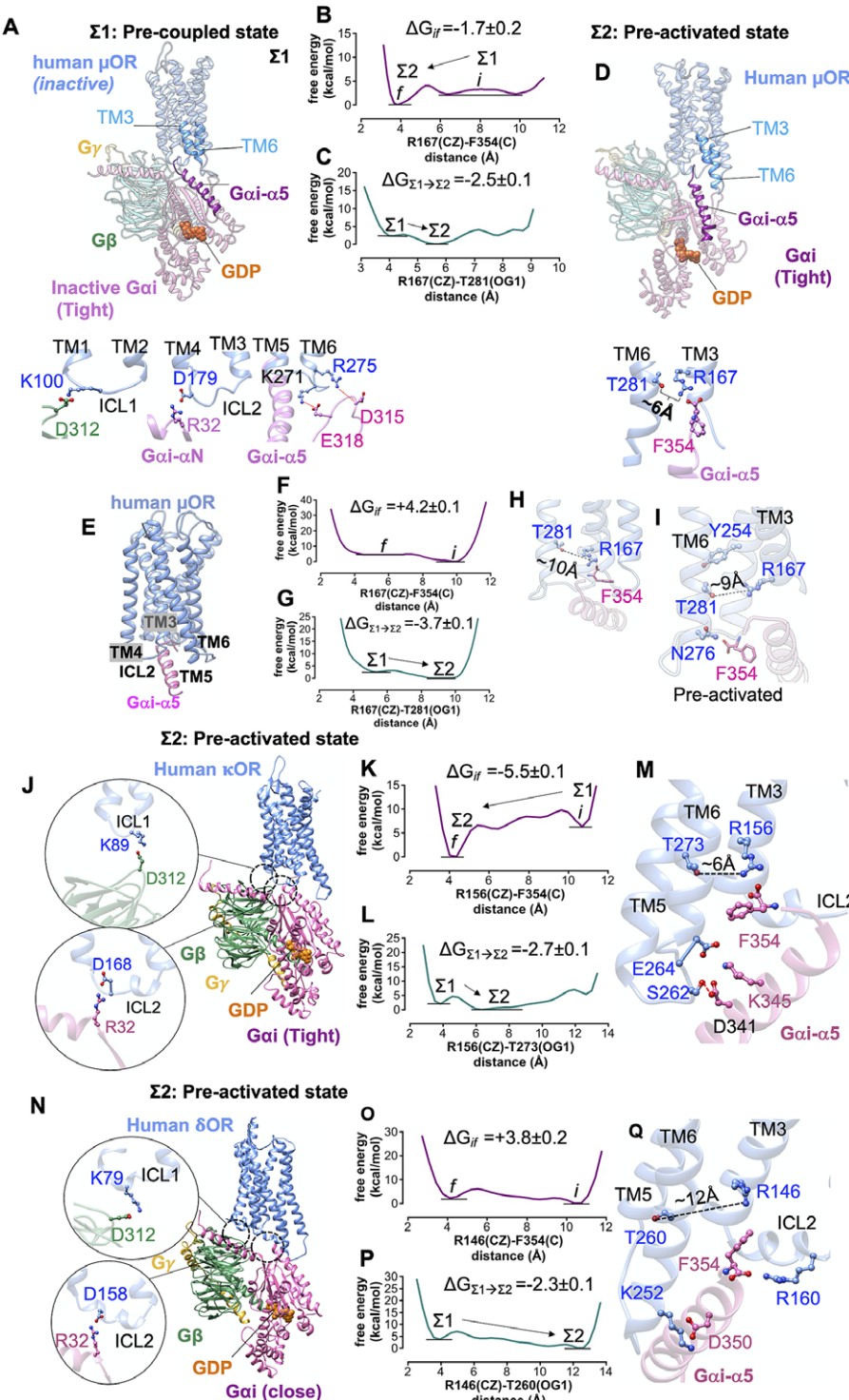

**Fig. 5.** Formation of pre-activated complex (Σ2) between Gi protein and opioid receptors prior to ligand binding. (*A*) The structure of the pre-coupled state (Σ1 state), comprising the inactive human μOR and inactive Gi protein-bound GDP (tight); the Gi protein binds to the inactive μOR by forming salt bridge anchors to ICL1, ICL2 and ICL3. (*B*) MetaMD free energy profile for the salt bridge between the F354 (C) and R167$^{3.50}$ (CZ). (*C*) MetaMD free energy for opening the polar interaction between R167$^{3.50}$ (CZ) and T281$^{6.34}$ (OG1) while F354 makes a strong salt bridge with R167$^{3.50}$. (*D*) The structure of the pre-activated state (Σ2) showing open μOR (broken polar interaction R167$^{3.50}$-T281$^{6.34}$), and the salt bridge between F354-R167$^{3.50}$. (*E*) The pre-activated complex (Σ2) between human μOR and the Gαi-α5 peptide (the rest of the Gi protein is eliminated) showing that formation of the pre-activated state is not dependent on a specific rigid-body orientation modelled in our Σ1 state. (*F*) MetaMD free energy profile for the interaction between the F354 (C) terminal carboxylate and R167$^{3.50}$ (CZ). (*G*) MetaMD free energy for breaking the polar interaction between R167$^{3.50}$ and T281$^{6.34}$. (*H*) The salt bridge between F354 (C) and R167$^{3.50}$ breaks the polar interaction between R167$^{3.50}$-T281$^{6.34}$. (*I*) Detailed structural analysis for the intracellular region of pre-activated complex between human μOR and the Gαi-α5 peptide. (*J,N*) The pre-activated complex between Gi protein and κOR/δOR, respectively. (*K,O*) MetaMD free energy profiles for the interaction between the F354 (C) and κOR-R156$^{3.50}$ (CZ) and δOR-R146$^{3.50}$ (CZ), respectively. (*L,P*) MetaMD free energy for breaking the coupling between TM3 and TM6 κOR: R156$^{3.50}$-T273$^{6.34}$ δOR: R146$^{3.50}$-T260$^{6.34}$. (*M,Q*) Detailed structural analysis for the intracellular region of pre-activated complex between human κOR/δOR and Gi protein. The weighted averages and the standard deviations were calculated for the converged period between the initial configuration before metaMD '*i*' and the final conformation→'*f*' after metaMD calculations (Supplementary Figs S4–S6).

specific rigid-body orientation of the Gi protein modelled in the Σ1 state.

To find if initiation of activation by Gi protein before agonist binding is statistically significant, we performed two independent metaMD simulations (an aggregate ~1.4 μs) on our model of the pre-coupled state. We followed the same molecular mechanism to characterise the pre-activated state of κOR-Gi and δOR-Gi. Thus, we placed the inactive Gi protein in close proximity of κOR and δOR so that it could form salt bridge anchors with the receptors. In addition, we used Charmm36m (Huang *et al.,* 2017) for these calculations to eliminate the possibility that the formation of pre-activated complex resulted solely from the choice of a specific force field.

Prior to agonist binding, the inactive Gi protein couples to inactive κOR and δOR to form a pre-activated complex (Fig. 5*j,n*) just as for the human μOR. The movement of Gαi-α5 helix into the inactive κOR (Fig. 5*k,l*) breaks the intrinsic polar interaction between R156$^{3.50}$ and T273$^{6.34}$ to open up space to accommodate the Gαi-α5 helix. Upon breaking the hydrogen bond between R156$^{3.50}$-T273$^{6.34}$, T273$^{6.34}$ rotates towards TM5, just as did the analogous T$^{6.34}$ in the μOR structure (Fig. 5*m*). Our calculations indicate that the hydrogen bond is broken because F354 forms a salt bridge with R156$^{3.50}$ (Fig. 5*k*). Similarly, the affinity between Gi protein and inactive δOR (Fig. 5*o,p*) breaks the polar interaction between R146$^{3.50}$ and T260$^{6.34}$ opening it to ~12 Å (Fig. 5*q*). We find that F354 makes a salt bridge with R146$^{3.50}$ (Supplementary Fig. S7), just as for μOR and κOR. But once the intracellular region of δOR opens up, F354 rearranges a polar interaction from its aromatic ring to the side chain of R146$^{3.50}$ (Fig. 5*q*). This allows the terminal carboxylate to establish a salt bridge with R160 on ICL2. Upon opening the polar interaction between R146$^{3.50}$-T260$^{6.34}$, T260$^{6.34}$ rotates towards TM5, a behaviour similar to the other opioid receptors. Thus, rotation of T$^{6.34}$ towards TM5 seems to be essential for the activation of opioid receptors.

### Completion of G protein activation by agonist binding

To determine the role of agonist in the G protein-first activation paradigm, we inserted the agonists to the pre-activated complex (Σ2) of opioid receptors: morphine in μOR, MP1104 in κOR and DPI-287 in δOR, to build the pre-activated complex bound to agonist (Σ3′). A salt bridge from conserved D$^{3.32}$ to the protonated N atom of agonists locks ligands into the orthosteric binding pocket of Σ3′ state (Supplementary Fig. S8).

We propose that agonist binding promotes the transformation of Σ3′ to Σ4′ by inducing the dramatic opening of the GDP binding pocket of the Gαi subunit. This opening of Gαi expedites GDP release, a critical event in activation of G protein and G protein signalling (Sprang, 1997; Oldham and Hamm, 2008). Thus, we examined the energetics of opening the AH and Ras-like domains that bind to the GDP (Fig. 5), using an aggregate ~1.1 μs metaMD simulations. Our analysis shows that once morphine, MP1104 and DPI-287 bind the μOR, κOR and δOR, respectively, the Gαi subunit undergoes a remarkable opening, separating the AH and Ras-like domains by ~16 to ~24 Å from the GDP binding site. This is energetically favourable (~−6 kcal mol$^{−1}$) in the presence of morphine (Fig. 6*a*) and leaves the GDP water exposed and susceptible to dissociation or GTP exchange. In fact, our independent metaMD simulations on unliganded-μOR complexed with the inactive Gi protein-bound GDP (Supplementary Figs S16 and S17) find that the activation of Gi protein (GDP) coupled to the opioid receptors prevails only in the presence of an agonist. Without the

agonist, opening the GαI subunit from the GDP binding pocket substantially increases the energy up to (~+30.0 kcal mol$^{−1}$, Supplementary Fig. S16).

We find that the Gαi opening in the presence of liganded-κOR/δOR requires overcoming an energy barrier of ~2 kcal mol$^{−1}$ to provide an exit pathway for GDP (Σ4′*, Fig. 6*b,c*). This dramatic change in the Gαi structure is essential to the later exchange of the GDP for a GTP and signalling. However, we find that GDP still has sufficiently high affinity to the Ras-like domain, to remain bound to Gαi. In the Σ4′* state, the GDP retains polar contacts to the Ras-like domain while breaking the polar interactions with the Gαi-AH domain. It is well-known that the Ras-like domain is sufficient for binding of nucleotides (Markby *et al.,* 1993). The Σ4′* state reveals that GDP release and exchange may not be rapid, which agrees with a previous observation (Dror *et al.,* 2015). Moreover, opening of Gαi from GDP binding site provides sufficient freedom to Gαi-α5 helix that it can penetrate deep into the accessible open intracellular region of opioid receptors to stabilise the fully active state. After GDP exchange or release, the Σ4′* eventually relaxes to the Σ4′ state as shown in Fig. 3.

In the ligand-first paradigm, the ability of the agonist to break open the TM3-TM6 coupling is crucial (Ballesteros *et al.,* 2001; Yao *et al.,* 2006). Thus, to examine if the binding of agonist can trigger the activation by opening the cytoplasmic region of μOR, we inserted the morphine into the extracellular binding portion of μOR such that the protonated amine moiety of morphine makes a salt bridge with D149$^{3.32}$ (Fig. 7*a*), locking the morphine in the binding pocket. Since the hallmark of GPCR activation is outward movement of the cytosolic end of TM6 to expand the intracellular cavity to accommodate the Gα-α5 helix, we performed an aggregate ~2.4 μs metaMD simulations to evaluate the energetics associated with this TM6 repositioning. We find that the optimised μOR-bound morphine adopts a closed cytoplasmic packing (Fig. 7*b*) that closely matches the crystallographic inactive μOR. Our analysis indicates that morphine bound μOR does not open the intracellular expansion; opening the distance between TM3 and TM6 (from ~10.5 to 18 Å) increases the energy by ~14 kcal mol$^{−1}$ (Fig. 7*c*). The tight cytoplasmic packing with the strong hydrogen bond (~−2 kcal mol$^{−1}$) between R167$^{3.50}$-T281$^{6.34}$ (Fig. 7*d,e*) impedes the TM6 from outward displacement. Thus, the binding of morphine does not shift the inactive state of μOR to an active conformation (Koehl *et al.,* 2018).

To examine if an agonist alone, in the absence of Gi protein or nanobody, could stabilise the active conformation of the μOR, we started with the fully active state of μOR bound to DAMGO and Gi protein resolved by Cryo-EM (Koehl *et al.,* 2018), and removed the Gi protein (Fig. 7*f*). Then we allowed the resulting μOR-DAMGO complex to equilibrate with an aggregate ~1.4 μs metaMD simulations. Contrary to general expectations, our free energy calculations (Fig. 7*g,h*) reveal that the TM6 undergoes a remarkable ~5 Å inward movement in an energetically downhill process, ~6 kcal mol$^{−1}$, contracting the intracellular cavity to reach the inactive crystallographic conformation (Fig. 7*i*). This contraction allows TM6 to couple to TM3 by making strong hydrogen bonds from T279$^{6.34}$ to R165$^{3.50}$ (Fig. 7*h*), the intrinsic characteristic of the inactive μOR. In a second study, we removed the nanobody from the active state of μOR bound to BU72 (Huang *et al.,* 2015) and carried out a ~960 ns metaMD simulation to allow the resulting μOR-BU72 complex (Fig. 7*j*) to equilibrate. Again, our free energy calculations find that TM6 moves towards TM3 by ~6 Å, with the energy decreasing by ~−2.2 kcal mol$^{−1}$, to convert the μOR from the activated structure to the crystallographic inactive conformation (Fig. 7*k–m*).

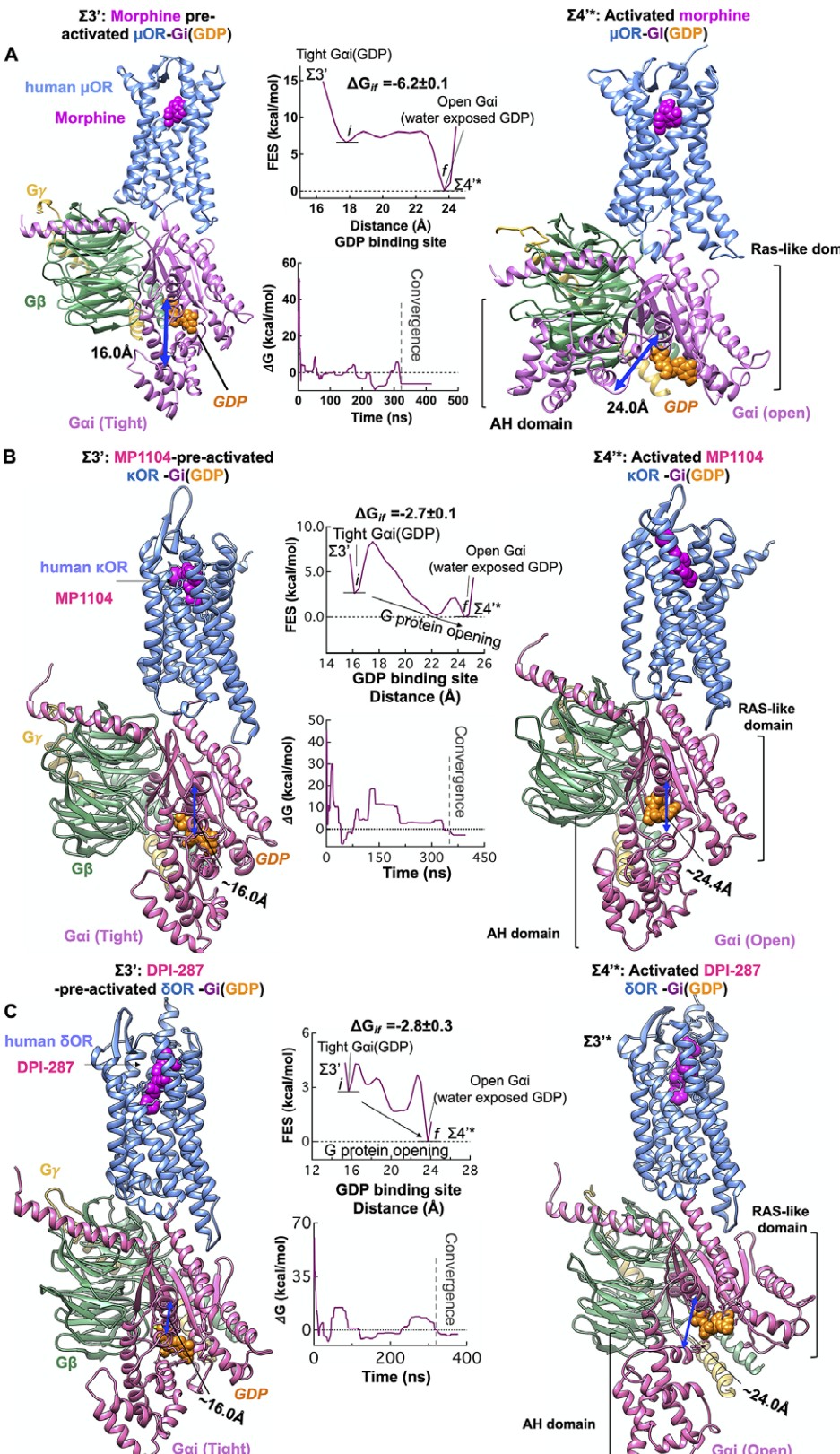

**Fig. 6.** Agonist promotes the activation of Gi protein by inducing opening of the Gαi subunit. Gi protein activation mediated by (*A*) morphine binding to pre-activated μOR-Gi protein complex, (*B*) MP1104 binding to pre-activated κOR-Gi protein complex and (*C*) DPI-287 binding to pre-activated δOR-Gi protein complex. Overall, we performed an aggregate ~1.1 μs metaMD simulations to evaluate the energetics relevant to opening the Gαi subunit from its GDP binding site. For these free energy calculations, the collective variable was the distance between the AH domain (the centre of mass of the Cα atoms for the residues 147–181) and the Ras-like domain (the centre of mass of the Cα atoms for the residues 42–59), which define the GDP binding site. The weighted averages and the standard deviations were calculated for the converged period between the initial configuration before metaMD '*i*' and the final conformation '*f*' after metaMD calculations.

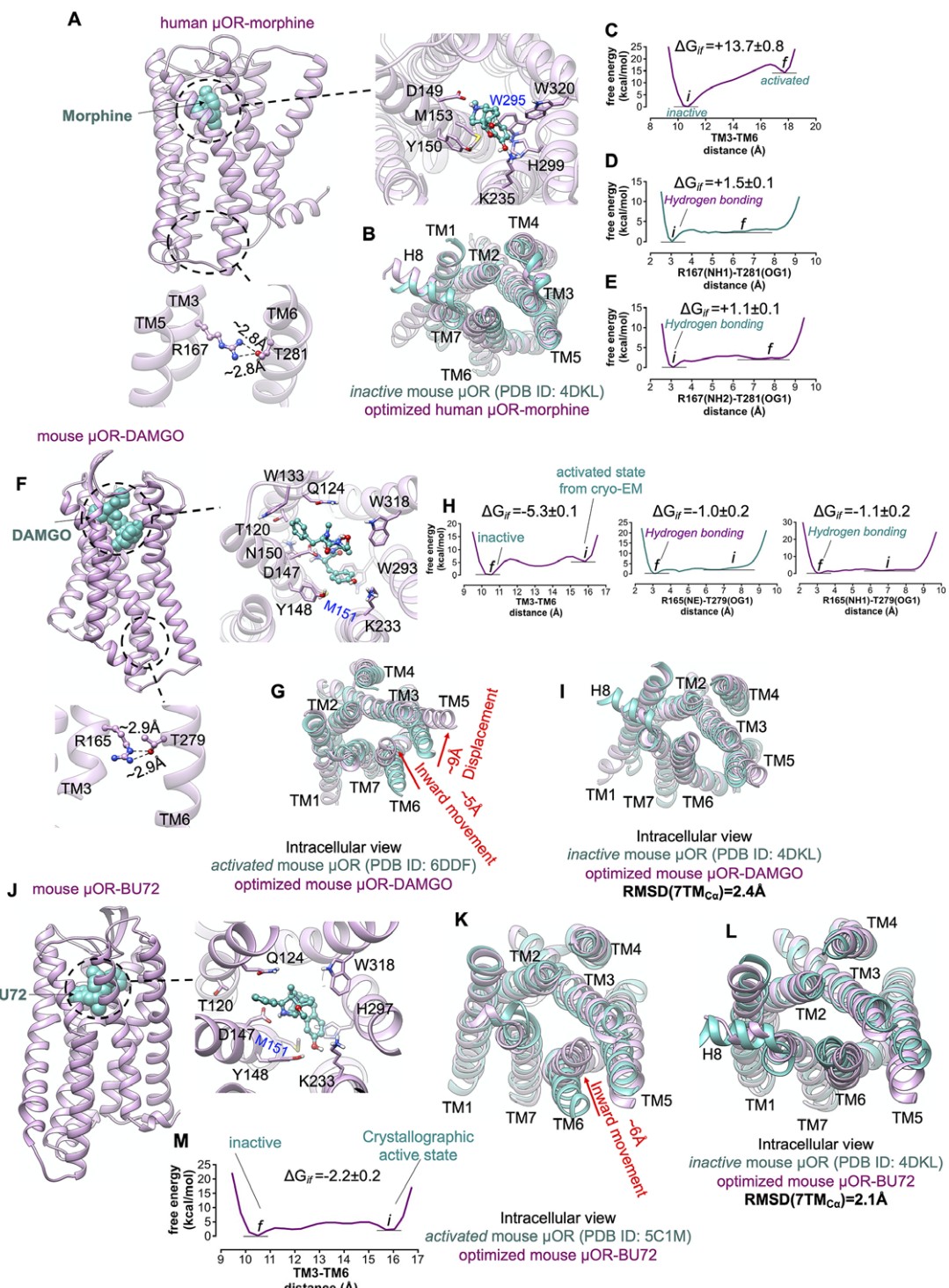

**Fig. 7.** μOR possesses a weak allosteric coupling. (*A,F,J*) Our minimised structures of human μOR-morphine, mouse μOR-DAMGO and mouse μOR-BU72, respectively, in the absence of Gi protein or a mimetic Gi protein nanobody. Agonists alone cannot open up the cytoplasmic region of μOR, especially the hydrogen bond between R$^{3.50}$ and T$^{6.34}$. (*B*) Comparison of the minimised human μOR-bound morphine (pink) with the crystallographic inactive conformation of mouse μOR (green) resolved by crystallography (Manglik *et al.*, 2012). MetaMD free energy of (*C*) the distance between TM3 (the centre of mass of Cα for residues 161–172) and TM6 (the centre of mass of Cα for residues 274–285). (*D*) The interaction between R167$^{3.50}$(NH1)-T281$^{6.34}$(OG1). (*E*) The interaction between R167$^{3.50}$ (NH2)-T281$^{6.34}$(OG1). (*G*) Comparison of the minimised mouse μOR-bound DAMGO (pink) with the crystallographic active state conformation of mouse μOR (green) resolved by Cryo-EM (Koehl *et al.*, 2018), which indicates that removing Gi protein from the fully active state leads to remarkable contraction in the cytoplasmic region of μOR. MetaMD free energy of (*H*) Left: the distance between TM3 (the centre of mass of Cα for residues 159–170) and TM6 (the centre of mass of Cα for residues 272–283), middle: the interaction between R167$^{3.50}$ (NE)-T281$^{6.34}$(OG1), right: the interaction between R167$^{3.50}$ (NH1)-T281$^{6.34}$(OG1). (*I*) Comparison of the minimised mouse μOR-bound DAMGO (pink) with the crystallographic inactive conformation of mouse μOR (green). Comparison of the minimised mouse μOR-bound BU72 (pink) with: (*K*) The crystallographic active state (Huang *et al.*, 2015) of mouse μOR (green). (*L*) The crystallographic inactive state of mouse μOR (green). BU72 alone (a nanobody removed from the complex) cannot maintain the active state conformation. (*M*) MetaMD free energy of the distance between TM3 (the centre of mass of Cα for residues 159–170) and TM6 (the centre of mass of Cα for residues 272–283). All RMSDs were calculated for the Cα atoms on the TM domains. The weighted averages and the SD were calculated for the converged period between the initial configuration before metaMD '*i*' and the final conformation '*f*' after metaMD calculations (Supplementary Fig. S9).

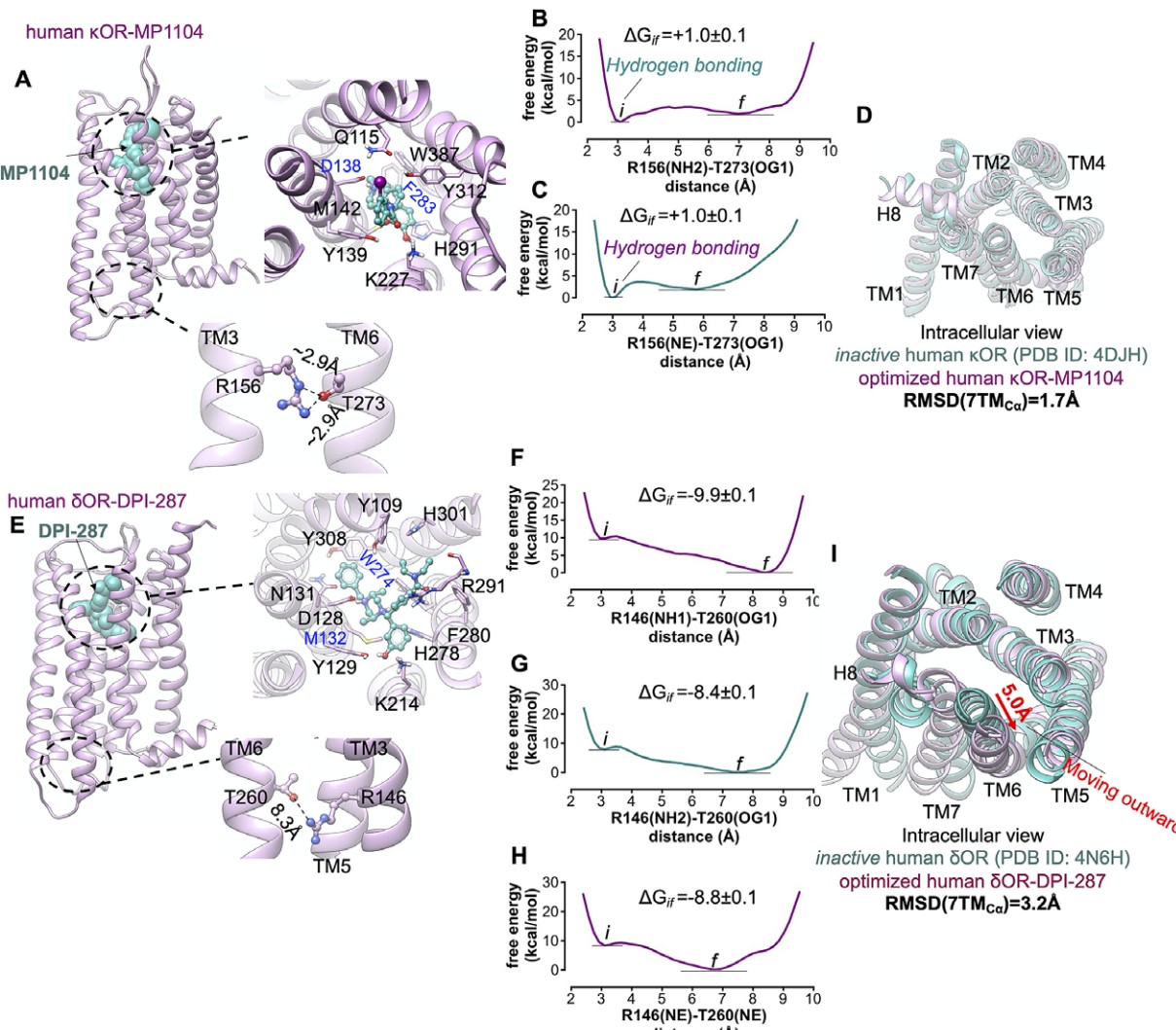

**Fig. 8.** Activation of Gi protein mediated by δOR favourably proceeds with both ligand-first and G protein-first activation mechanisms. (*A,E*) Our minimised structures of human κOR-MP1104 and human δOR-DPI-287, respectively, in the absence of Gi protein. MetaMD free energy of (*B*) the interaction between R156$^{3.50}$ (NH2)-T273$^{6.34}$(OG1) and (*C*) the interaction between R156$^{3.50}$ (NE)-T273$^{6.34}$(OG1). (*D*) Comparison of the minimised human κOR-MP1104 (pink) with the crystallographic inactive conformation of human κOR (green) resolved by crystallography (Wu *et al.,* 2012). MetaMD free energy of (*F*) the interaction between R146$^{3.50}$ (NH1)-T260$^{6.34}$(OG1), (*G*) the interaction between R146$^{3.50}$ (NH2)-T273$^{6.34}$(OG1) and (*H*) the interaction between R146$^{3.50}$ (NE)-T260$^{6.34}$(OG1). (*I*) Comparison of the minimised human δOR-DPI-287 (pink) with the crystallographic inactive conformation of human δOR (green) resolved by crystallography (Fenalti *et al.,* 2014). All RMSDs were calculated for the Cα atoms on the TM domains. Variation of the free energy difference with time was monitored to evaluate the convergence of metaMD simulations. The weighted averages and the standard deviations were calculated for the converged period between the initial configuration before metaMD '*i*' and the final conformation '*f*' after metaMD calculations (Supplementary Fig. S10).

Our free energy calculations show that μOR features a loose allosteric coupling between the ligand-binding pocket and the Gi protein coupling interface which is consistent with the previous nucleic magnetic resonance study (Sounier *et al.,* 2015), revealing that BU72 does not stabilise the active conformation of μOR in the absence of downstream proteins.

To determine whether agonist binding to inactive conformation of κOR and δOR opens up the TM3-TM6 polar interaction, we inserted MP1104 and DPI-287 to the binding site of κOR and δOR, respectively, where the protonated N atom of agonists makes a salt bridge with D$^{3.32}$ (Fig. 8*a,e*). We allowed these GPCR-bound agonist structures to equilibrate by performing metaMD simulations. We find that MP1104 fails to break the hydrogen bond between R156$^{3.50}$-T273$^{6.34}$ (Fig. 8*b,c*), which impedes the intercellular region from expansion, with the cytoplasmic configuration remaining close to the inactive state (Fig. 8*d*). Thus, κOR possesses a weak allosteric coupling.

In contrast to κOR and μOR, we find that the binding of DPI-287 to δOR can spontaneously break open the hydrogen bond between R146$^{3.50}$-T260$^{6.34}$ to ~7 Å (~−9 kcal mol$^{-1}$), allowing TM6 to experience a 5 Å outward movement (Fig. 8*f–i*), which expands the intracellular cavity to recruit Gi protein. This outward movement is a hallmark of activation in Class A GPCRs (Hilger *et al.,* 2020). Thus, agonist binding to δOR can indeed shift the inactive to the active conformation, encouraging Gi protein activation. As a result, activation of Gi protein mediated by δOR favourably proceeds with either ligand-first or G protein-first activation mechanisms.

## Discussion

We have shown that agonist binding to the pre-activated state of the Gi protein-opioid receptor completes the activation process

triggered by the initial binding of Gi protein. Our free energy calculations confirm that the G protein-first activation mechanism provides a sequence of thermodynamically favourable events that lead to activation of opioid receptors and Gi protein. Indeed, we expect that the most active agonists must bind strongly to the pre-activated structure and that they must lead to a small barrier to induce the $\Sigma3'$ state to open the closed G$\alpha$ while releasing the GDP to progress towards the 'activated' structure ($\Sigma4'$) described above.

A very important implication of our new G protein-first activation mechanism is that for a ligand to activate the G protein, it must bind to the pre-activated state, $\Sigma2$, forming $\Sigma3'$, which then must open up the AH and Ras-like subdomains of G$\alpha$ tightly coupled to the GDP to form the final fully activated open G$\alpha$ with the AH and Ras-like subdomains widely separated, as observed experimentally, $\Sigma4'$. The binding of agonists to pre-activated state is likely different than the final binding site in $\Sigma4'$ observed experimentally that must be considered in designing agonists. Indeed, we found important differences in the pharmacophore of $\Sigma3'$ compared to $\Sigma4'$. Unfortunately, the structure for an agonist bound to $\Sigma2$ to form $\Sigma3'$ has not yet been observed experimentally. Our only knowledge of this structure is from the simulations.

A second important consideration is that the agonist binding in $\Sigma3'$ must facilitate opening of the tightly coupled AH and Ras-like subdomains of G$\alpha$ couple tightly to the GDP into the final fully activated open G$\alpha$ observed experimentally. It is likely that the barrier for this activation may depend sensitively on the structure of the agonist and the binding site. It may be that a full agonist has a low barrier, but a partial agonist may have a higher barrier or two different barriers depending on the structure of the agonist. Thus, uncovering the G protein-first mechanism for G protein activation is just the first step in gaining control over the activation processes.

## Methods

We performed long-scale MD and metaMD simulations for an aggregate ~21 μs as described in detail in the Supplementary Information to characterise the activation pathway of opioid receptors in accord with the G protein-first mechanism of activation using multitude of available Cryo-EM and crystal structures. We used three well-known force fields of Amber14 (Dickson *et al.,* 2014), Charmm36m (Huang *et al.,* 2017) and OPLS (Robertson *et al.,* 2015) to exclude the possible impacts of applied force field on results.

For the free energy calculations, the temperature was maintained at 310 K using a velocity-rescale (Bussi *et al.,* 2007) thermostat with a damping constant of 1.0 ps and the pressure was controlled at 1 bar using a Parrinello–Rahman barostat algorithm (Parrinello and Rahman, 1981) with a 5.0 ps damping constant. Semi-isotropic pressure coupling was used during this calculation. The Lennard–Jones cutoff radius was 10 Å, where, the interaction was smoothly shifted to 0 after 10 Å. Periodic boundary conditions were applied to all three directions. The Particle Mesh Ewald algorithm (Essmann *et al.,* 1995) with a real cutoff radius of 10 Å and a grid spacing of 1.2 Å was used to calculate the long-range coulombic interactions. A compressibility of $4.5 \times 10^{-5}$ bar$^{-1}$ was used in the *xy*-plane and also the *z* axis, to relax the box volume. In all the above simulations, water OH-bonds were constrained by the SETTLE algorithm (Miyamoto and Kollman, 1992). The remaining H bonds were constrained using the P-LINCS algorithm (Hess, 2008). All simulations were performed using GROMACS (Pronk *et al.,* 2013; Abraham *et al.,* 2015) and free energy calculations were done using PLUMED-2 (Tribello *et al.,* 2014).

**Acknowledgements.** We thank Dr. Sijia Dong and Dr. Fan Liu for helpful discussions. This project was initiated with support by the GIST-Caltech Collaboration with Prof. Yong-Chul Kim of GIST, Korea. It was completed with support from NIH (R01HL155532). These calculations used the computational resources funded by DURIP (ONR N00014-16-1-2901) and the XSEDE (Extreme Science and Engineering Discovery Environment) supported by National Science Foundation Grant (ACI-1548562).

**Supplementary Materials.** To view supplementary material for this article, please visit http://dx.doi.org/10.1017/qrd.2021.7.

**Author contributions.** W.A.G. and A.M. designed the project. A.M. carried out all calculations. A.M. and S.-K.K. prepared all figures and the Supplementary Information. W.A.G., A.M. and S.-K.K. wrote the manuscript.

**Conflict of interest.** The authors declare no conflicts of interest.

**Open Peer Review.** To view the open peer review materials for this article, please visit http://dx.doi.org/10.1017/qrd.2021.7.

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
