## [Reviewer Report]

*Comments to Author*: This is an interesting study of the activation of opiod receptors, with a focus on differences between the G-protein coupled receptors μOR, κOR, and δOR regarding their interaction with the Gi protein, differences that may help in the design of drugs with reduced side effects.

The authors use model building based on available Cryo-EM and X-ray structures to create receptor-Gi complex structures that are subjected to extensive (a total of 17 μs) molecular dynamics simulations. The results strongly suggest that in order for the receptor to activate the Gi protein the agonist first has to bind to the receptor.

In the metadynamics simulations used to characterize the strength of specific interactions, the quantitative end result is sensitive to the choice of reaction coordinate (or collective variable). Qualitatively the observed differences are likely to be reliable indicators of the differences between the systems.

---

## [Reviewer Report]

*Comments to Author*: The manuscript presents an impressive exploratory work on different phases of the mechanism of activation of the complex between human opioid receptors, Gi proteins and agonists, using extensive meta-molecular dynamics simulation. Activation involve large conformational changes in the opioid receptor as well as the Gi protein complex. One question the study aims at enlightening concerns the order in which the partnersactto trigger allosteric responses in the opioid receptor or in the Gi protein. Notably, the "in silico" observations are confronted to two possible mechanisms, the Ligand-First or the G Protein-First mechanisms. To test whether observations arising from the simulation represent general properties of opioid receptors, and whether they are robust, the authors reproduce the study on three closely related receptors and use three different force fields. The work involves modeling unknown forms of opioid receptor substates, notably based on homology or combining parts of the system with known structure. It also involves free energy calculations of different transition pathways within the receptors or the Gi proteins. Overall, this is a very consequent and convincing work, that reveals new elements of the mechanism such as the pre-positioning and pre-anchoring of the G-protein to the inactive opioid receptor via strong interactions with the ICL loops, and provides interpretation for known mutational effects in the μOR receptor (e.g. R181C).Discussion of the Ligand-First or the G Protein-First mechanisms elaborates on the simulation results and the observation that one of the studied receptors, δOR, presents conformational change properties that differ from the other two. Finally, the gained knowledge has strong importance for therapeutical applications in pain-relief treatments that would avoid side effects.

The manuscript is clearly written and presents the results of a a large body of carefully executed and analyzed integrative work. Questions and comments below intend to help further clarifying the results and conclusions of this dense study of a very complex system, since there is space for improving the understanding for a general audience that is non-necessarily specialist of the question.

A. Main comments.

1) The authors study several steps of the activation process, named from Σ0 to Σ4', but it is not always clear in the main text which step, or the transition between which steps, is studied. It would be useful for the comprehension that the names of the steps be linked to the different substate models that are studied. I am conscious that since the mechanism is still not well established, an unequivocal correspondence between the mechanistic steps and the conformational substates may not yet be possible. In that case, I recommend the authors name the conformational substates they study using different labels (e.g. A, B, C…), that they describe the characteristics of each substate (i.e. inactive/active receptor; inactive/active G protein, no/with ligand) and they explicit the possible routes the activation process may follow along these substates in the Ligand-Fist or G Protein-First mechanisms, together with the possible correspondence with the Σ steps. A scheme would be very useful in this purpose, that the reader could relate to in the course of the reading instead of struggling with activity shades such as "fully active", "active", "pre-activated", "inactive" that do not directly tell which component is in which form.

2) The authors performed very long simulations aiming at ensuring correct equilibration. However, when it comes to sampling the optimal binding geometry (p11 "…allowed the pre-coupled complex to equilibrate by performing a ~1μs metaMD simulation allowing the complex to find the optimum position and orientation of Gαi-α5"), even a μs metadynamics sampling may be insufficient if another favorable position exists in the space perpendicular to the sampled coordinate, that would be separated by a high energy barrier. The authors address this possibility by sampling the position of the isolated Gαi-α5 helix, again using metaMD (p.12).They write that the helix explored "various positions and orientations" but without quantifying how much "varied" the sampling happened to be. Could the authors quantify the sampling? Why did not the authors use methods that sample discrete positions and orientations on the receptor surface, independent from each other, such as macromolecular docking methods, in order to pre-identify possible geometries of interaction that could further be thoroughly sampled via metaMD? This would ensure that all possible positions and orientations have been considered.

3) Role of the ligand:the authors showed that at least for two receptors, the presence of the ligand alone is not sufficient for the transition to the receptor active state (with separated TM3 and TM6 helices) to occur. In another work cited in the manuscript (Huang et al., Nature 2015), it was proposed that a cation occupies the binding site in addition to the ligand; the authors of this other work did add a positive charge during molecular dynamics simulations they performed on the system. Was a charge added in the present simulation? May an added charge modify the observed results about the transition of the receptor to an active state?

4) (a) I did not understand whether the conformation of the receptor in the sigma2 state is the same as the conformation of the receptor in the fully active state, or if the binding of the ligand further changes the conformation of the receptor: could the authors provide this precision ? (b) The authors write that the presence of the ligand induces a large conformational change in the Gi protein; however, unless I missed something (see comment 1), it seems that the transition of the Gi protein to its the fully active state is only explored in the presence of the ligand: would it be possible to have the same exploration done in the absence of the ligand but with the receptor in its active state in order to understand what exact role the ligand plays ?I could not understand the exact role of the ligand based on the present description; (c) the authors refer several times to a "weak coupling" (e.g. p.21: "Thus, κOR possesses a weak allosteric coupling")? In which way do their observation support the existence of a coupling between the presence of the ligand and the Gi protein conformational change?

5) I am not sure whether the present calculations definitely speak for the G protein-first mechanism in the case of the mu- and kappa-ORs: what the authors clearly showed is that the ligand alone cannot turn an inactive state of the receptor towards an active one, nor can they stabilize the active state and impede it to return to an inactive state in the absence of the Gi protein. However, the authors also write "We showed that this critical event of activation is common to both Ligand-First and G Protein-First mechanisms of activation". In fact, what I understand of their argument in support of the G protein-First mechanism is that the Gi protein binding is ruled by collisions and is therefore slow, while "we expect that agonists will contact opioid receptors that are already pre-activated by the tight Gi protein, making ligand activation independent from slow random collisions." Is agonist binding accelerated by the fact that the receptor is in a pre-activated state? Are there arguments in favor of the agonist binding process not being based on slow random collisions? And again, how exactly does ligand activation function (question 4)? I recommend the authors explain their point more clearly and separate arguments that arise from their present results from what arguments taken from the literature.

B. Minor comments

1) Figures: I recommend labelling helix α5 of Gαi, helices TM3 and TM6 as well as the GDP, the Ras-like and the AH domains, in at least one of the figures (e.g. Fig.1); these structural elements are essential for the activation pathway, therefore it would be useful to easily locate them in the models. In addition, the extensive direct use of Ballesteros-Weinstein numbering for GPCRs in some places make make it difficult for non-specialists to follow the results description (e.g. p 6, "E28^( GαiN)-R182⁴.⁴⁰": which protein does R182⁴.⁴⁰ belong to?).

2) Please, indicate in the main text (in both the Results and the Methods sections) that the full description of the modeling process can be found in supplementary information. The methods section is very short and little informative, this is all right if the reader is referred to the supplementary information but this needs to be plainly written.

3) p.8: while the authors fully convinced me they reached a higher resolution state than the Cryo-EM structure, and notably that they revealed the highly probable existence of anchoring interactions between the receptor and the G protein, I do not understand the interest of building a density map from the results of ~ 1 μs MD simulation with restrained backbone atoms. Clearly, this map better correlates with the protein coordinates solved by CryoEM than the CryoEM map itself, but the MD map was created directly from the protein coordinates solved by Cryo-EM. The fact that the map is "closer" to the coordinates may simply reflect the fact that the MD sampling was not sufficient (notably because of the restrained backbone atoms).Could the authors better justify their argument?

4) p.8, and p.1 SI, the authors use the mouse operator as a template to generate the active human form and to position the morphine ligand. What is the degree of sequence similarity between the mouse and the human receptors ? specifically, what is the degree of sequence similarity of the morphine binding sites?

5) p.8 "We discovered that the Gi protein interfaces the human μOR by forming salt bridge anchors to ICL1, ICL2, ICL3, and the cytoplasmic end of TM6.tOur analysis shows that the Gβ subunit makes a direct and stable ionic contact from D312Gβ to K100ICL1 (Figure 2B & 2H).": While the described investigations and the comparison with other systems are very convincing in favor of the existence of an interaction network between the ICLs and the Gi protein, one cannot be 100% sure that an alternative interaction network is not possible. I would prefer that the authors use a formulation that still leaves place for uncertainty, even while keeping the word "discovery" (which is justified by the discovery of a highly probable and yet unknown interaction network between the ICLs and the G protein).

6) p.10 : "Prior to the ligand binding, Gi protein has sufficient time to couple to the opioid receptors to form a pre-coupled state.": could the authors provide some support to this affirmation ?

7) p11 "Subsequently, allowed the pre-coupled complex to equilibrate by performing a ~1μs metaMD simulation allowing the complex to find the optimum position and orientation of Gαi-α5.":"we" allowed is probably missing.

8) p.12, the formation of a salt bridge contact between the Gαi-α5 helix and the receptor is presented as a trigger to weaken, and then break the interaction between helices TM3 and TM6. Could the authors show a plot with the chronology of α5-receptor salt bridge formation/TM3-TM6 interaction breaking in support to their description ?

9) (a) Fig.4 & Fig.5 Could the authors provide additional explanation about what do these relevant free energy minima represent and why the passage between them need to be damped? I am not sure I understand that point, and the mention of the two minima treatment somewhat blurs the main description.

(b) In a general way, I think most of the caption of Figure 5 should be displaced to the Material and Methods section, or to the protocol description in supplementary information.

---

## [Reviewer Report]

*Comments to Author*: Reviewer #2: This is an interesting study of the activation of opiod receptors, with a focus on differences between the G-protein coupled receptors μOR, κOR, and δOR regarding their interaction with the Gi protein, differences that may help in the design of drugs with reduced side effects.

The authors use model building based on available Cryo-EM and X-ray structures to create receptor-Gi complex structures that are subjected to extensive (a total of 17 μs) molecular dynamics simulations. The results strongly suggest that in order for the receptor to activate the Gi protein the agonist first has to bind to the receptor.

In the metadynamics simulations used to characterize the strength of specific interactions, the quantitative end result is sensitive to the choice of reaction coordinate (or collective variable). Qualitatively the observed differences are likely to be reliable indicators of the differences between the systems.

Reviewer #3: The manuscript presents an impressive exploratory work on different phases of the mechanism of activation of the complex between human opioid receptors, Gi proteins and agonists, using extensive meta-molecular dynamics simulation. Activation involve large conformational changes in the opioid receptor as well as the Gi protein complex. One question the study aims at enlightening concerns the order in which the partnersactto trigger allosteric responses in the opioid receptor or in the Gi protein. Notably, the "in silico" observations are confronted to two possible mechanisms, the Ligand-First or the G Protein-First mechanisms. To test whether observations arising from the simulation represent general properties of opioid receptors, and whether they are robust, the authors reproduce the study on three closely related receptors and use three different force fields. The work involves modeling unknown forms of opioid receptor substates, notably based on homology or combining parts of the system with known structure. It also involves free energy calculations of different transition pathways within the receptors or the Gi proteins. Overall, this is a very consequent and convincing work, that reveals new elements of the mechanism such as the pre-positioning and pre-anchoring of the G-protein to the inactive opioid receptor via strong interactions with the ICL loops, and provides interpretation for known mutational effects in the μOR receptor (e.g. R181C).Discussion of the Ligand-First or the G Protein-First mechanisms elaborates on the simulation results and the observation that one of the studied receptors, δOR, presents conformational change properties that differ from the other two. Finally, the gained knowledge has strong importance for therapeutical applications in pain-relief treatments that would avoid side effects.

The manuscript is clearly written and presents the results of a a large body of carefully executed and analyzed integrative work. Questions and comments below intend to help further clarifying the results and conclusions of this dense study of a very complex system, since there is space for improving the understanding for a general audience that is non-necessarily specialist of the question.

A. Main comments.

1) The authors study several steps of the activation process, named from Σ0 to Σ4', but it is not always clear in the main text which step, or the transition between which steps, is studied. It would be useful for the comprehension that the names of the steps be linked to the different substate models that are studied. I am conscious that since the mechanism is still not well established, an unequivocal correspondence between the mechanistic steps and the conformational substates may not yet be possible. In that case, I recommend the authors name the conformational substates they study using different labels (e.g. A, B, C…), that they describe the characteristics of each substate (i.e. inactive/active receptor; inactive/active G protein, no/with ligand) and they explicit the possible routes the activation process may follow along these substates in the Ligand-Fist or G Protein-First mechanisms, together with the possible correspondence with the Σ steps. A scheme would be very useful in this purpose, that the reader could relate to in the course of the reading instead of struggling with activity shades such as "fully active", "active", "pre-activated", "inactive" that do not directly tell which component is in which form.

2) The authors performed very long simulations aiming at ensuring correct equilibration. However, when it comes to sampling the optimal binding geometry (p11 "…allowed the pre-coupled complex to equilibrate by performing a ~1μs metaMD simulation allowing the complex to find the optimum position and orientation of Gαi-α5"), even a μs metadynamics sampling may be insufficient if another favorable position exists in the space perpendicular to the sampled coordinate, that would be separated by a high energy barrier. The authors address this possibility by sampling the position of the isolated Gαi-α5 helix, again using metaMD (p.12).They write that the helix explored "various positions and orientations" but without quantifying how much "varied" the sampling happened to be. Could the authors quantify the sampling? Why did not the authors use methods that sample discrete positions and orientations on the receptor surface, independent from each other, such as macromolecular docking methods, in order to pre-identify possible geometries of interaction that could further be thoroughly sampled via metaMD? This would ensure that all possible positions and orientations have been considered.

3) Role of the ligand:the authors showed that at least for two receptors, the presence of the ligand alone is not sufficient for the transition to the receptor active state (with separated TM3 and TM6 helices) to occur. In another work cited in the manuscript (Huang et al., Nature 2015), it was proposed that a cation occupies the binding site in addition to the ligand; the authors of this other work did add a positive charge during molecular dynamics simulations they performed on the system. Was a charge added in the present simulation? May an added charge modify the observed results about the transition of the receptor to an active state?

4) (a) I did not understand whether the conformation of the receptor in the sigma2 state is the same as the conformation of the receptor in the fully active state, or if the binding of the ligand further changes the conformation of the receptor: could the authors provide this precision ? (b) The authors write that the presence of the ligand induces a large conformational change in the Gi protein; however, unless I missed something (see comment 1), it seems that the transition of the Gi protein to its the fully active state is only explored in the presence of the ligand: would it be possible to have the same exploration done in the absence of the ligand but with the receptor in its active state in order to understand what exact role the ligand plays ?I could not understand the exact role of the ligand based on the present description; (c) the authors refer several times to a "weak coupling" (e.g. p.21: "Thus, κOR possesses a weak allosteric coupling")? In which way do their observation support the existence of a coupling between the presence of the ligand and the Gi protein conformational change?

5) I am not sure whether the present calculations definitely speak for the G protein-first mechanism in the case of the mu- and kappa-ORs: what the authors clearly showed is that the ligand alone cannot turn an inactive state of the receptor towards an active one, nor can they stabilize the active state and impede it to return to an inactive state in the absence of the Gi protein. However, the authors also write "We showed that this critical event of activation is common to both Ligand-First and G Protein-First mechanisms of activation". In fact, what I understand of their argument in support of the G protein-First mechanism is that the Gi protein binding is ruled by collisions and is therefore slow, while "we expect that agonists will contact opioid receptors that are already pre-activated by the tight Gi protein, making ligand activation independent from slow random collisions." Is agonist binding accelerated by the fact that the receptor is in a pre-activated state? Are there arguments in favor of the agonist binding process not being based on slow random collisions? And again, how exactly does ligand activation function (question 4)? I recommend the authors explain their point more clearly and separate arguments that arise from their present results from what arguments taken from the literature.

B. Minor comments

1) Figures: I recommend labelling helix α5 of Gαi, helices TM3 and TM6 as well as the GDP, the Ras-like and the AH domains, in at least one of the figures (e.g. Fig.1); these structural elements are essential for the activation pathway, therefore it would be useful to easily locate them in the models. In addition, the extensive direct use of Ballesteros-Weinstein numbering for GPCRs in some places make make it difficult for non-specialists to follow the results description (e.g. p 6, "E28^( GαiN)-R182⁴.⁴⁰": which protein does R182⁴.⁴⁰ belong to?).

2) Please, indicate in the main text (in both the Results and the Methods sections) that the full description of the modeling process can be found in supplementary information. The methods section is very short and little informative, this is all right if the reader is referred to the supplementary information but this needs to be plainly written.

3) p.8: while the authors fully convinced me they reached a higher resolution state than the Cryo-EM structure, and notably that they revealed the highly probable existence of anchoring interactions between the receptor and the G protein, I do not understand the interest of building a density map from the results of ~ 1 μs MD simulation with restrained backbone atoms. Clearly, this map better correlates with the protein coordinates solved by CryoEM than the CryoEM map itself, but the MD map was created directly from the protein coordinates solved by Cryo-EM. The fact that the map is "closer" to the coordinates may simply reflect the fact that the MD sampling was not sufficient (notably because of the restrained backbone atoms).Could the authors better justify their argument?

4) p.8, and p.1 SI, the authors use the mouse operator as a template to generate the active human form and to position the morphine ligand. What is the degree of sequence similarity between the mouse and the human receptors ? specifically, what is the degree of sequence similarity of the morphine binding sites?

5) p.8 "We discovered that the Gi protein interfaces the human μOR by forming salt bridge anchors to ICL1, ICL2, ICL3, and the cytoplasmic end of TM6.tOur analysis shows that the Gβ subunit makes a direct and stable ionic contact from D312Gβ to K100ICL1 (Figure 2B & 2H).": While the described investigations and the comparison with other systems are very convincing in favor of the existence of an interaction network between the ICLs and the Gi protein, one cannot be 100% sure that an alternative interaction network is not possible. I would prefer that the authors use a formulation that still leaves place for uncertainty, even while keeping the word "discovery" (which is justified by the discovery of a highly probable and yet unknown interaction network between the ICLs and the G protein).

6) p.10 : "Prior to the ligand binding, Gi protein has sufficient time to couple to the opioid receptors to form a pre-coupled state.": could the authors provide some support to this affirmation ?

7) p11 "Subsequently, allowed the pre-coupled complex to equilibrate by performing a ~1μs metaMD simulation allowing the complex to find the optimum position and orientation of Gαi-α5.":"we" allowed is probably missing.

8) p.12, the formation of a salt bridge contact between the Gαi-α5 helix and the receptor is presented as a trigger to weaken, and then break the interaction between helices TM3 and TM6. Could the authors show a plot with the chronology of α5-receptor salt bridge formation/TM3-TM6 interaction breaking in support to their description ?

9) (a) Fig.4 & Fig.5 Could the authors provide additional explanation about what do these relevant free energy minima represent and why the passage between them need to be damped? I am not sure I understand that point, and the mention of the two minima treatment somewhat blurs the main description.

(b) In a general way, I think most of the caption of Figure 5 should be displaced to the Material and Methods section, or to the protocol description in supplementary information.